# Functional Mesoporous Silica Nanocomposites: Biomedical Applications and Biosafety

**DOI:** 10.3390/ijms20040929

**Published:** 2019-02-20

**Authors:** Rafael R. Castillo, María Vallet-Regí

**Affiliations:** 1Dpto. Química en Ciencias Farmacéuticas. Facultad de Farmacia, Universidad Complutense de Madrid, Plaza Ramón y Cajal s/n, 28040 Madrid, Spain; rafcas01@ucm.es; 2Centro de Investigación Biomédica en Red—CIBER, 28029 Madrid, Spain; 3Instituto de Investigación Sanitaria Hospital 12 de Octubre—imas12, 28041 Madrid, Spain

**Keywords:** mesoporous silica, magnetic, photothermal, photodynamic, combination therapy, drug delivery

## Abstract

The rise and development of nanotechnology has enabled the creation of a wide number of systems with new and advantageous features to treat cancer. However, in many cases, the lone application of these new nanotherapeutics has proven not to be enough to achieve acceptable therapeutic efficacies. Hence, to avoid these limitations, the scientific community has embarked on the development of single formulations capable of combining functionalities. Among all possible components, silica—either solid or mesoporous—has become of importance as connecting and coating material for these new-generation therapeutic nanodevices. In the present review, the most recent examples of fully inorganic silica-based functional composites are visited, paying particular attention to those with potential biomedical applicability. Additionally, some highlights will be given with respect to their possible biosafety issues based on their chemical composition.

## 1. Introduction

The unstoppable advance of nanotechnology during the last decades has led to the development of a large number of nanomaterials with great therapeutic potential. Focusing on the treatment of cancer, the biomedical area with the greatest penetration of nanometric-based therapies, it is possible to find a large number of materials suitable for also treating a large number of pathologies [1,2]. For example, magnetic [3,4] and plasmonic materials [5,6,7] could be remotely excited with magnetic fields and light, respectively to produce a thermal response capable of inducing cell apoptosis. Similarly, photodynamic chemical sensitizers [7] can generate Reactive Oxygen Species (ROS), inducing cell death throughout the oxidative stress pathway. Although these alternative therapies can improve the prognosis of the disease when employed as adjuvants, the truth is that chemotherapy remains the reference treatment. 

Fortunately, delivery of chemotherapeutics has also evolved, allowing the development of nanoencapsulated drug formulations able to improve the pharmacological profile of free drugs. In the clinic, liposomes [8,9] stand out, distantly followed by drug-protein hybrids and polymeric particles [1]. Nonetheless, mesoporous silica (MS)-based materials are slowly beginning to gain relevance because of two unique properties: their well-established drug delivery properties [10,11] and their versatility for creating high-performing hybrid materials, which is the purpose of the present review. The classic features of Mesoporous Silica Nanoparticles (MSNs), related to their porous and robust structure, have given them notoriety in the academic field as drug delivery systems [12]. However, in the last decade, the implementation of silica technology with other nanomaterials has allowed the successful combination of several components in single nanometric entities. This, apart from being of academic interest, has also aroused a certain clinical interest, since the additional nanosystems’ functionality could be successfully integrated with the well-known drug load capabilities of the MS. Therefore, the multiple integration of features in single entities could be a pipe dream, with its definitive weakness being that, in the same way that raw nanomaterials have certain security issues and limitations, these are transferred to composite materials. In this review, we will focus on the development of hybrid inorganic-silica materials with multiple features with applicability in nanomedicine and cancer treatment. Moreover, the additional features and known safety issues related to each functional silica-containing composites will be discussed; however, as the MS will be common to all materials, we will begin with the biosafety aspects of silica. Apart from MS, there have also been reported other multifunctional devices based on different nanoplatforms [13,14,15,16].

Bulk silica is widely present in food additives and cosmetic products, which indicates low toxicity in the body; in general, silica is catalogued as “Generally Recognized as Safe” by the FDA (US Food and Drug administration, ID Code: 14808-60-7). Despite the demonstrated biocompatibility of bulk silica, when any chemical species is prepared as a nanomaterial, new risks, limitations and safety issues appear as a consequence of their tiny size. For example, the intimate interaction between particles and cells that allow nanomaterials to be assimilated by tumor cells, could generate unexpected side effects in healthy organs if they are accumulated or induce sensitization [17].

Despite these limitations, MSNs are also considered to be safe materials for biomedical applications [18,19,20], as they suffer from slow hydrolysis in aqueous media [21,22,23], which may last up to weeks depending on pore conformation and functionalization. For raw MSNs, matrix degradation is a phenomenon of capital importance to properly determining the release kinetics; although in functional composites and modified silica, this parameter is harder to control, as more parameters apply and much less information is available.

Apart from the degradability and the possibility to be excreted (Figure 1), there are other parameters and effects that should be accounted on MSNs in order to prepare them as nanomedical devices [23,24]. Size [25], shape [26,27,28], pore volume [29] and surface functionalization are also capital aspects. Regarding morphology, it is accepted that MSNs between 100 and 200 nm are the best performing particles; as they (1) avoid fast clearance [20] and acute toxicity [30,31], which are associated with small particles, and (2) avoid aggregation on physiological fluids, blood capillaries and alveoli [32], as associated with bigger particles. Moreover, rod-like particles seem to behave better than spherical ones thanks to the fact that multivalent interactions with membranes are easier [26,33].

Regardless of the morphology, the superficial modification of the MSNs plays a fundamental role in their efficacy and safety. First of all, it must be borne in mind that the use of the EPR effect alone, although it could improve the behavior of the free drugs, is often not enough to give rise to personalized therapies and to improve the efficiency of available treatments. For this reason, several nanosystems have been created that are able to increase specificity towards certain cell populations. In this sense, a multitude of systems in which silica is coupled with recognition elements such as antibodies, aptamers and other bioactive fragments have been described [35,36,37]. On the other hand, it is also important to account for the fact that exogenous nanoparticles undergo a deposition of serum proteins as a result of the interaction with the immune system. This process, called opsonization, leads to a strong surface modification—a protein corona [38,39]—which produces disruption of the particles’ recognition abilities, leading finally to their elimination. Fortunately, opsonization could be mitigated by including highly hydrophilic polymers or fragments such as polyehtyleneglycol [40,41] or zwitterion structures able to create strong hydration layers on the surface [42,43].

In addition to surface modification, it is also important to consider that connections between components require additional in-between layers. One of the most common substances for such purposes is dense silica, usually employed in thicknesses of around several nanometers. The role of this intermediate layer usually goes beyond providing chemical inertness and magneto-optical transparency. It also permits the components to be physically separated, avoiding physicochemical processes like dissolution or passivation of the internal core and the quenching of fluorescent-labelled composites. Moreover, it also permits the generation of additional mesoporous layers without adding complexity to the system. It is also important to note that these dense silica layers have demonstrated additional features, some of them reviewed below, such as increased photothermal stability of cores or the possibility to tune relaxitivity of contrast agents in Magnetic Resonance Imaging [44,45,46]. In view of all of these aspects, to currently classify MSNs as suitable devices for biomedical applications, they must comply with series of morphology and surface modification requirements aimed at minimizing the immune response—stealthing—and enhancing tissue/cell recognition—targeting. As expected, all these features must be also compiled if silica-containing inorganic composites are intended for biomedical applications. 

## 2. Inorganic-Mesoporous Silica Nanocomposites with Magnetic Response

### 2.1. Magnetic Materials in the Clinic

The use of magnetic nanomaterials is one of the most active research areas in biomedical research [47], as proven by the large number of these nanosystems undergoing clinical trials [2]. Usually, these magnetic materials consist of crystalline iron oxides whose surface has been engineered to avoid agglomeration and accelerated dissolution [48]. Among the known nanometric magnetic materials, Superparamagnetic Iron Oxide Nanoparticles (SPIONs) are by far the most developed systems [49]. In this way, SPIONs are magnetized in the presence of external magnetic fields, while they do not show such behavior in its absence. Moreover, SPIONs show a diverging behavior depending on the external magnetic field. Hence, if constant magnetic fields are applied (permanent magnet), magnetized SPIONs could be remotely guided towards the magnet, while if under alternating magnetic fields, SPIONs experience magnetization shifts. If this oscillation is powerful enough, the continuous reorientation of the magnetization would generate a magnetic resistance which would be thermally dissipated. This effect, when employed on living systems, leads to heat-induced apoptosis, known as magnetic hyperthermia [50]. Additionally, SPIONs have broad applicability as contrast agents for Magnetic Resonance Imaging (MRI) [51,52], since they can be easily modified to tune their affinity to cells or tissues.

Regarding the biosafety of clinically interesting SPIONs, it is important to remark that iron oxide crystals smaller than 20 nm suffer from quick clearance, but outstanding biocompatibility [53], and require surface coating to obtain higher circulation times and immune stealthing. Among all coatings, dextran, starch, or PEG polymers are the most employed, with the low molecular weight of dextran being the most recurrent in clinical trials [2,50].

### 2.2. Iron Oxide-Mesoporous Silica Nanocomposites (IOMSNs) as Drug Delivery Agents

The coating of raw SPIONs is of interest because it makes is possible to increase time-stability by preventing its dissolution and undesired aggregation processes, among other important features. Among all SPION-silica hybrid species, the mesoporous ones have special relevance, as they make it possible to develop Magnetic Drug Delivery Systems (MDDS). Nevertheless, this coating may also induce use limitations, as a MS shell may produce thermal insulation of the magnetic core [54]. This effect, although with little relevance in MRI and magnetic guidance, needs to be accounted for if Magnetothermal-chemotherapy (MTCT) is intended [55]. Despite the improved in vivo stability provided by the silica coating of IOMSNs, this outer shell must also be chemically modified to avoid the interaction with adhesive proteins of the immune system [56]. This could be achieved by any of the conventional coating procedures known for silica: polyethyleneglycol, [40,57] small targeting elements, functional polymers, zwitterions, small-interfering RNAs (siRNAs), deposition and coating with membranes, although none of these materials have yet reached clinical studies, as far as we know (Figure 2). Among all of the surface coatings developed so far, PEG is by far the most frequently recurring component for SPIONs and IOMSNs [58]. However, if such MDDS are not properly targeted, they will show a limited cell uptake and incremented off-target accumulation, which may lead to severe problems of misplaced heating. To avoid this, much effort has been devoted to the development of PEG-modified polymers able to efficiently target cells.

In addition to PEG-based polymers, there are many other interesting examples of polymer-coated IOMSN composites in the literature. For example, Zhang and coworkers employed a poly dopamine pH-sensitive shell to retain Doxorubicin (DOX) within the mesopores [59]. Additionally, this system was functionalized with a folate-modified PEG to provide targeting towards HeLa breast cancer cells. The efficacy of this coating was assessed using cell viability studies, which demonstrated high survival rates in vitro when the coating was present. The folate targeting employed made it possible to reach high values of tumor accumulation without significant accumulation in major organs. In another example by Hanagata, a thermosensitive polymeric shell was designed for IOMSNs [60]. Thereby, the chosen *N*-isopropylacrylamide (NIPAM) containing polymer had the ability to change its conformation from globular to expanded when heated. With this system the authors could efficiently deliver DOX in combination with magnetic induced heating. This system proved to have an enhanced apoptotic effect on HeLa cells. Moreover, Guisasola et al. [61] employed an equivalent device in vivo. In this work, it could be seen that combination of hyperthermia and chemotherapy led to tumor growth reversion, demonstrating the potential of MTCT.

Another common polymer employed in MDDS is polyethyleneimine. This is widely employed because of its cationic character, which permits the bridging together of negatively charged moieties like particles and nucleic acids. For example, Lee and coworkers developed a multilayered Zn-doped IOMSN magnetic device that was able to successfully deliver DOX and the lethal-7a (let-7a) micro RNA (miRNA) able to disrupt Heat-Shock Protein expression. To do so, the authors prepared negatively charged phosphonate IOMSNs, which were coated with PEI, (10 kDa) polymer. Then, onto it was deposited the miRNA and an RGD-modified PEI layer [62]. The resulting RGD-targeted cationic assembly provided enhanced internalization on MDA-MB-231 breast cancer cells. This MDDS provided a satisfactory tumor volume reduction, although magnetic hyperthermia was not employed.

Although polymer coating of raw and hybrid MSNs enables outstanding retention of drugs within the mesopores, it is necessary to note that those polymers usually have associated severe drawbacks due to their limited degradability and/or intrinsic toxicity. For example, PEI polymers larger than 25 kDa [63] are known to highly destabilize cell membranes, promoting apoptosis, and unreacted NIPAM monomers are highly toxic [64], while PEG polymers generate a growing immunity [65,66] due to their ubiquity in pharmaceuticals. For these reasons, many research groups are trying to avoid these issues by moving towards the use of more convenient biogenic elements to develop targeting and gating. For example, Popova et al. employed chitosan and alginate polymers to make a multilayer coating of sulfonate modified IOMSNs [67]. This double coating is based on the electrostatic interaction of the cationic chitosan with both anionic IOMSNs and the alginate layer. This particular assembly permitted loading two different drugs: mitoxantrone within the mesopores, and prednisolone in between the chitosan-alginate layer. Another example was reported by Sinha et al., who deposited dextran onto boronic acid-modified IOMSNs [68]. This system proved to be sensitive to glucose, which was able to displace the dextran chains from the boronic acid. This system proved to successfully deliver Camptothecin on HeLa breast cancer cells thanks to the presence of folic acid as targeting agent. Unfortunately, the potential of this device was not fully tested, as the combined apoptotic effect of MTCT was not addressed. Glycopolymers have also been employed for coating IOMSNs; for example, Lactobionic-2-aminoethyl methacrylate and 3-(Methacryloxy)-propyltrimethoxysilane were co-polymerized onto IOMSNs for drug delivery [69]. In this case, the pH-sensitive polymeric shell was able to retain DOX, unless a lysosomal escape occurred. Although the potential for MTCT was not addressed, the authors determined the fate of these MDDS in a murine model employing MRI detection.

To enhance composites’ performance, another exploited approach is the modification of the surface with targeting elements. In an interesting contribution by Gao et al., folic acid was employed as a coating for the silica layer [70] to develop a targeted system able to exert MTCT. In this system, as a thermo-responsive nanogate was not included, the DOX release showed the same pattern in the presence or absence of the alternating magnetic field. As expected, the application of a simultaneous MTCT showed a synergistic apoptotic effect in vitro on human MCF-7 breast cancer cells. Similarly, a recent paper by Kariduraganavar and coworkers employed IOMSNs with transferrin on its surface to address U87 human glioblastoma cells [71]. Moreover, the authors employed a Blood-Brain Barrier model to demonstrate that these nanosystems are able to cross this barrier.

Apart from biogenic moieties, many functional modifications have also been included in IOMSNs. For example, Portilho et al. designed a theranostic system using a surface ^99^Tc-containing ligand for SPECT imaging with dacarbazine for the detection and treatment of melanoma [72]. Another synthetic approach, aimed at improving circulation time, is the use of zwitterion elements on the surface. These molecules, which have perfectly balanced but separate charges, are known to create a strong hydration layer that prevents protein adhesion [73]. In a nice example by Sánchez-Salcedo et al., phosphonate-capped IOMSNs were coated with a low-weight cross-linked PEI, onto which was grafted a zwitterion phosphorylcholine fragment [74]. Moreover, in this nanosystem, the remaining free amino groups from the PEI were balanced with a siRNA to achieve multimodal magnetically induced MTCT plus gene silencing. For the time being, this system has only been partially tested in vitro, with promising results on Ovcar8 ovarian cancer cell line.

In addition to polymers and small molecules, another promising coating for IOMSNs is lipid bilayers [75], similar to the so-called protocells. In a pioneering example by Mattingly et al., a cationic lipid shell was employed to entrap DOX within IOMSNs [76]; unfortunately, this magnetic protocell led to relevant death rates on tested cells due to the cationic lipid bilayer employed. Nevertheless, this example set the basis for Sen and coworkers to design a phospholipid-coated system to deliver DOX to MCF-7 and U87 cell lines [77]. In this case, the authors obtained an outstanding biocompatibility, unless drugs or magnetic hyperthermia were applied. More recent examples deepened the potential of this capping strategy, as elegantly demonstrated by Li and coworkers, who cloaked their nanocomposite with red blood cell membranes [78] to obtain excellent biocompatibilities and circulation times (Figure 3). 

In addition to organic and biological coatings, other interesting systems with promising behavior have been also developed. For instance, Liu et al. coated folate-targeted IOMSNs with a fully biocompatible CaCO_3_ as an acid-cleavable gatekeeper for Daunorubicin delivery [79]. This strategy, although it profits from a simple and effective end-cap coating method based on the precipitation of CaCO_3_ from CaCl_2_ and Na_2_CO_3_, seems to affect the targeting capabilities of the complete system, as suggested by the loss of preferential uptake.

### 2.3. Magnetic Composites with Applications in Nuclear Magnetic Imaging (MRI)

The use of superparamagnetic materials in MRI is of great importance to achieve better contrast between tissues [51]. In the case of transversal magnetization imaging (*T2* weighted MRI, employed mainly for low-fat highly hydrated tissues), SPIONs are known to shorten the spin−spin relaxation time of water and to provide better contrast in lean tissues such as liver, spleen and kidneys. On the other hand, longitudinal magnetization imaging (*T1* weighted MRI, more useful to visualize tissues with high fat or low water contents) is improved when paramagnetic cations—Mn^2+^, Gd^3+^ and mainly Fe^3+^—are used as sensitizers.

Unlike sensitizers exclusively designed for MRI, nanodevices containing metal and silica oxides are gaining interest, as they can be employed for simultaneous diagnosis and therapy—theranosis [80,81]. Moreover, the sensitivity provided by IOMSNs for *T2*-MRI is usually higher than that obtained for SPIONs, as recurrently demonstrated in the literature [82,83]. This effect could also be obtained by increasing the number of sensitizers through strategies that employ a greater number of metal oxides, which would be generally placed at the outermost layer of the composite. These Mesoporous Silica-Metal Oxide Nanocomposites (MSMONs) show promising properties because of the higher metal loads achieved and the outstanding biocompatibility of the components. Nevertheless, this approach also has limitations, as coating nanoparticles must be functionalized too to avoid any undesired aggregation processes, opsonization, off-target uptake and/or immune response.

The coating of MSNs with SPIONs was first reported by Victor Lin’s group, who designed a stimulus-responsive delivery system using silica nanorods [84]. Although this responsive composite was intended as a proof of concept, the authors clearly outlined a future biomedical approach as they employed a redox-cleavable linker sensitive to biogenic compounds. More recently, Hyeon and coworkers developed PEG-coated MSMONs employing chemical ligation between amino-capped MSNs and bromoalkyl-modified SPIONs [85]. To do so, they PEGylated the outer surface following a two-step methodology: (1) reacting the amino groups with a succinimidyl carboxylate PEG, which was then (2) functionalized with previously PEGylated SPIONs. This model exemplifies the difficulty of preparing MSMONs, which, apart from needing two different stealthing agents, must be carefully synthetized to avoid massive aggregation between particles.

With the previous design in mind, and with the aim of developing a synchronous *T1* MRI contrast agent, Zou and coworkers developed a MSMON employing Ultra-small Manganese Oxide Nanoparticles to cap the mesopores [86]. Those caps could quickly dissolve under weakly acidic conditions and release Mn^+2^ to enable *T1* weighted imaging. Moreover, the authors employed DOX within the pores to have an additional therapeutic effect (Figure 4). Employing a similar strategy, Huang et al. designed a system in which MSNs were doped with Fe^3+^ and loaded with DOX [87]. This device proved to release ferric cations, together with DOX, when mild-acidic environments were encountered, accounting its use as a theranostic platform too. Moreover, although not reported, these DOX-loaded, metal-doped MSNs are supposed to be fully biodegradable, as only Si, Mn and Fe oxides are employed in their synthesis (Figure 4).

Regarding Gd, the most widely employed contrast agent for MRI, there have also been reported a broad number of systems. Like in the previous examples, the strongly paramagnetic Gd^3+^ ions could be located either on the surface of MSNs through known chelants, doped within the porous silica matrix, or even as a core-shell structure. The first approach, chelation throughout ligands, proved to be suitable for the generation of MSNs with contrast properties for MRI [88,89]. Along this line, it is important to note the work by Davis and coworkers, who determined that surface location of Gd-chelates led to better contrast and sensitivity [90]. Unfortunately, this approach has a severe drawback, as surface modification is highly limited and complex due to the presence of Gd-chelates.

For this reason, new strategies have been developed for the incorporation of Gd into functional nanosystems. One of those possibilities is doping the silica matrix, which liberates the surface for further functionalization. However, a poor signal-to-noise ratio arises as a consequence of placing Gd in a mismatched crystallographic matrix. Hence, to overcome this issue, two strategies that employ compact Gd-matrices have been reported. The first, in which the Gd occupies the core and the silica the shell, has the advantage of enabling multimodal detection throughout *T1* weighted MRI and NIR-emitting persistent luminescence. Moreover, such an approach makes it possible to further modify the MS layer with all the developed targeting and stealthing technology, as elegantly demonstrated by the Yu and Chen groups [91,92]. The other possibility for obtaining solid Gd-containing matrices is based on the construction of Gd-shells [93], although this strategy does not profit from the advantages associated with mesoporous silica coatings. The previous examples use MSNs to, upon thermal treatment, dope the silica matrix. This strategy, although suitable for the release of paramagnetic cations, does not enhance the SPION-mediated *T2* imaging. To solve this problem, Miao and coworkers devised a strategy in which MSNs were coated with core-shell SPIONs@MnO nanoparticles [94]. This system could be disassembled when either (1) pH is below 5, (2) Glutathione (GSH) is overexpressed, or (3) ROS are above normal values. As expected, such systems showed negligible effect on the viability unless Camptothecin was loaded, thus demonstrating outstanding biocompatibility.

Regarding the biosafety of these species, it should be noted that capping nanoparticles must comply with all the premises made at the beginning of this revision. They must have an adequate surface functionalization to provide enough colloidal stability and to avoid the action of the immune system. Moreover, nanocaps must be either efficiently cleared or biodegraded; although in any case, use of small nanoparticles for capping purposes clearly aids in their final elimination. Regarding the cores, the use of ionic species (such as iron oxides reviewed herein or CuS and UCNs reviewed below) does not seem to be a problem regarding bioaccumulation as they degrade; although they may suffer from acute toxicity issues. However, the use of non-degradable species (as Au) must be carefully addressed, as will be discussed below.

### 2.4. Remote Homing of Magnetic Mesoporous Silica Nanocomposites

One of the most acclaimed applications of MDDS is the possibility of achieving magnetic guidance, although for magnetic silica composites, it is difficult to find successful examples. One of those is the model reported by Lee, which profited from a facilitated in vitro magnetic transfection to increase the apoptotic effect induced by hyperthermia plus DOX and let-7a miRNA co-delivery into glioblastoma multiform cancer cells [62]. 

However, when magnetic guidance is intended in vivo, the round-shaped composites do not seem to be the choice option. For these purposes, bifunctional nanobullets consisting of a Fe_3_O_4_ magnetic head attached to a mesoporous silica tail seem to be a better option, according to the results published by Dong and coworkers [95,96,97]. On their first formulation, the authors employed Fe_3_O_4_ magnetic particles produced throughout high-temperature hydrolysis with poly(acrylic acid) as surfactant. Then, MS rods were prepared onto those by controlled addition of the silica precursor. The resulting MS tail was functionalized with amino groups, onto which PEG was grafted, obtaining the desired *Janus* Iron Oxide Mesoporous Silica Nanobullets [95]. This system was widely tested in vitro, demonstrating very nice biocompatibility, together with evidence of energy-dependent clathrin uptake.

In vivo bioaccumulation of this system in HepG2 tumor-bearing mice demonstrated a preferred accumulation on the spleen. However, when magnetically targeted, the liver and the hepatic tumor were the tissues with the highest accumulation, proving an effective magnetic guidance. Unluckily, despite the obtained tumor accumulation, the use of these nanobullets as DOX delivery agents did not improve the use of free DOX. To overcome this negative result, in a subsequent investigation the authors implemented the system by using a prodrug (Ganciclovir) loaded in the mesopores, plus the herpes simplex virus thymidine kinase (prodrug activator) outside the pores. To retain the enzyme within the carrier, the system was finally coated with a poly(l-lysine)-poly(ethylene glycol) layer grafted onto the carboxylate-modified nanobullets throughout amide coupling [96]. Both in vivo and in vitro studies showed that combined guidance and MTCT made it possible to significantly reduce the viability of cells and tumor sizes. More recently, these authors continued to study the possibilities of this bullet-like MDDS, in this case, to deliver curcumin to HepG2 cells [97]. The result of combining these three effects—magnetic induced delivery, hyperthermia and curcumin delivery—was an effective tumor reduction, which was not obtained when any of these three proapoptotic factors were absent.

In addition to this, in these works, the authors also tackled four different but important aspects of MDDS: (1) they compared the round-shaped (isotropic) core-shell disposition against the nanobullet (anisotropic) configuration, finding higher drug loading and faster release for the nanobullets; (2) the bullet-like materials demonstrated better magnetic guidance and hyperthermia due to their non-insulated magnetic head; (3) nanobullets provided an enhanced gene delivery when compared to the core-shell IOMSNs; and finally (4) proved that the use of hyperthermia in combination with the gene suicide therapy led to significant tumor growth control.

In summary, Dong’s group demonstrated the versatility of their nanobullets platform for multifunctional nanosystems, as they successfully accomplished remote guidance, drug delivery, hyperthermia and MRI. Moreover, the use of stealth PEG-based coatings made it possible to reduce aggregation and immune system clearance thus favoring diffusion and increasing the therapeutic effect. As a drawback, the authors have not yet addressed the use of efficiently targeted nanobullets, which might improve the efficacy of the magnetic-induced accumulation (Figure 5).

### 2.5. Magnetic-Based Nano- and Biosensors

Magnetic materials have another important field of development in nanosensors. One of the most promising approaches, although restricted to in vitro assays, is Magnetic Relaxation Switching [98]. This strategy profits from the increase of *T2* transversal relaxation time suffered by SPIONs when aggregated, which, if caused by an analyte, would enable the development of MRI- [99] or relaxometry [100]-based sensors. However, this approach is not suitable for most of the systems described herein, as the MS layer does not permit such close contact aggregation.

In addition to them, magnetic nanomaterials have also been described as lipophilic sample enrichment systems, which are based on the interaction of hydrophobic-coated SPIONs with lipophilic substances. Please refer to the following reviews to deepen understanding on the uses of hydrophobic (C8 to C18 chains, Carbon nanotubes, Graphene, Surfactants, Polymers, etc.) modified IOMSNs [101,102]. From a safety point of view, although these nanosystems are able to adsorb substantial amounts of highly lipophilic proteins and analytes, the nature of the surface modifications makes these systems extremely incompatible with living organisms and restricts their use to in vitro purposes only. 

### 2.6. Magnetic Biostimulators

Another important aspect of magnetic therapy is the remote induction of latent behaviors in treated cells. Among these, the most exploited is hyperthermia, an apoptosis inductor; however, new interesting possibilities for other bone diseases have also been reported [103]. Along these lines, Deng and coworkers employed IOMSNs arrays to suppress osteoclast differentiation [104]. This effect opened the way to developing novel therapies for osteoporosis, mainly when in combination with the antiresorptive effect of zoledronic acid. To prepare such arrays, the authors prepared silica-coated SPIONs. Then, those were aligned with an external magnetic field, maintaining the alignment thanks to a magnetic dipolar interaction; then, it was possible to coat the array with an additional mesoporous layer. The resulting linear, stable mesostructures could be prepared with lengths of up to 15 μm. The considerable length showed by this particular composite material made it possible to remotely induce shear forces, which proved to inhibit osteoclast differentiation in concentrations above of 62.5 ng/mL. However, despite the significant advance represented by potential remote disruption of the resorptive pathway for osteoporosis, the biosafety of this material could be questionable; first, the enormous (from 1 to 15 μm) length might induce severe embolisms due to composite aggregation, mainly after considering that the outer silica layer is not treated with stealthing agents. However, additionally, the delivery of those materials towards the target could also be problematic due to their complicated delivery to the bone. All their limitations notwithstanding, the evolution of these IOMSNs’ nanochains is of extraordinary interest, as it represents the first remotely triggered example for treating osteoporosis.

## 3. Inorganic-Mesoporous Silica Nanocomposites Responsive to Light

### 3.1. Nanomaterials with Thermochemical Response: Photothermal and Photodynamic Effects

It is well known that certain materials are able to interact with light-producing physical effects different from refraction, absorption and dispersion. In some cases, the light stimulation may produce either a thermal or an electronic excitation, which could be of interest for biomedical applications. In this way, if the resulting effect is thermal, it will be considered as a photothermal effect (PTE) [105], while if it is a light-driven chemical excitation, we would talk about a photodynamic effect (PDE) [106,107]. This second effect is originated when the electronic excitation is relaxed by an energy-transfer to surrounding molecules, leading to ROS that may undergo oxidative stress in living systems. The third possibility considered herein is luminescence, which will be reviewed below. All these possibilities have proved their feasibility for biomedical applications, thanks to several highly interesting contributions, although in some cases, their use is not exactly innocuous.

The photothermal effect can be achieved either with organic or inorganic materials. The organic ones include the use of carbon allotropes [108] (graphene, nanotubes, etc.) and conjugated polymeric materials [109]; whereas among the inorganics, the greatest exponents are those derived from Au [5,110]. The common problem for all of these materials is their enormous propensity to accumulate in living organisms, as they are not easily biodegraded. This can lead to both long-term toxic effects and latent, but unwanted, off-target photothermal effects. For this reason, it is of vital importance to use these photoactive species in sizes that allow their efficient excretion, and in very small quantities. Fortunately, the development of composite materials with a MS coating makes it possible to tune the overall size, avoiding their aggregation. Moreover, the use of silica for PTE is highly convenient, because of its transparency in near infrared (NIR), although this results in the thermal insulation of the photothermal core, and thus lower response. In addition to therapeutic PTE, NIR excitation could also be employed for on-demand drug delivery purposes; for more information on this topic, please refer to reference [111]. 

PDE occurs when the sensitizer is able to gather photons and transfer them to other reactive chemical species. Photosensitizers (PS) could be of either organic or organometallic nature—phenothiazinium cations, porphyrins, and phthalocyanines [112,113]—but also of inorganic nature. Among these, examples of particles with PDE employing Fe_3_O_4_ [114], Ti_2_O [115] and ZnO [116,117] nanoparticles have been reported. Moreover, PDE is also common for Au and CuS nanoparticles as a side relaxation pathway complementary to PTE. The use of PDE with Fe, Ti and Zn oxides is based on UV irradiation and has severe limitations for living systems. Fortunately, as will be discussed below, the use of UV radiation can be avoided by using composite materials containing upconversion particles, which are able to gather photons from low-energy radiation and convert them onto UV radiation able to excite photosensitizers and thus create the desired ROS. The main advantage of this strategy is the use of a harmless light source and the extremely localized generation of UV light, which only occurs where upconversion nanoparticles are present. These two aspects enhance the biosafety of composites for PDE, although only if the PS is eliminable. For more information about PDE, please check the outstanding review authored by Lucky, Soo and Zhang [107].

In addition to their potential, non-degradable photosensitizers have severe side effects, including latent sensitivity, off-target accumulation, and chronic toxicity. For this reason, the scientific community has been interested in the development of safer alternatives. Relevant examples for PTE induction could be achieved with degradable CuS [118,119] and Fe_3_O_4_ [120,121] crystals, which, due to their ionic nature, could be fully biodegraded. Unfortunately, to the best of our knowledge, there are no low risk photosensitizer alternatives for PDE. Their side-toxicity or latent reactivity must be always accounted for in biomedical applications. An in-depth discussion of the advances that provide the combination of photodynamic or photothermal therapies, together with anticancer drugs, can be found in a previous review by us. Interested readers, please check reference [55] for details. 

### 3.2. Inorganic Mesoporous Silica Nanocomposites for Photothermal Therapy

As previously introduced, the most typical materials for PTE are Au species, and among them Gold Nanorods (GNRs). Nevertheless, they show two important limitations: the impossibility of being degraded, and the presence of remainder surface surfactant CTAB hauled from its synthesis. Fortunately, the toxic cationic surfactant issue could be mitigated by coating the GNRs with a MS matrix. The resulting Mesoporous Silica Coated Gold Nanorods (MSGNRs) provide a stable coating that prevents aggregation and morphology shifts, while preserving its plasmonic properties and thermal response (Figure 6). In a pioneering example, Wu, Chen and coworkers were the first to deliver DOX with MSGNRs [122], proving that the simultaneous photothermal-chemotherapy (PTCT) provided an enhanced apoptotic effect. Inspired by the potential of PTCT, many other examples were reported in the following years, profiting from the facile surface functionalization of the MS outer layer. Those modifications could be aimed at modifying the release profile of loaded drugs or to turn the systems into actively targeted devices.

With respect to drug release modification, many examples can be found in the literature. For example, the system designed by Lin, Qu and coworkers employed pH-sensitive imine bonds to connect DOX and MS, obtaining a pH-driven release [123]. On the other hand, there have also been reported targeted systems. In one of the first examples, Liu et al. reported the surface modification of MSGNRs with a PEG polymer modified with a targeting tLip-1 peptide selective to the Neurophilin, a protein present in many cancer cell lines [124].

Capped systems have also been an object of study, although in this case, the interest is shifted to thermally responsive systems more than for PTCT. In one of the previous examples, Duget and coworkers coated folate-targeted MSGNRs with a low-temperature (39 °C) melting biocompatible component [125]. The 1-tetradecanol employed made it possible to effectively coat the outer MS layer, thus allowing the retention of small molecules within the mesopores. However, when the system was exposed to NIR light and heated, the gatekeeper melted, and the drug could be released. In this case, the authors did not pay attention to possible sensitivities to the 1-tetradecanol release, which may have disrupted the normal function of membranes and enhanced the overall apoptotic effect. On this same topic, our group developed a reversible nanosystem in which pore gating was accomplished by a NIPAM-based thermo-sensitive polymer [126]. To obtain appropriate functionalization and an adequate transition temperature, a hydroxyl-containing monomer was included. Onto it, there could be anchored a bifunctional PEG, which made it possible to incorporate the melanoma-targeting NAPamide peptide. The system, evaluated in vitro, proved to be selectively accumulated to melanoma cells, producing the expected increase of cell death when combined PTCT was exerted.

Apart from AuNRs, there are other Au species that show plasmonic properties. Among them, nanostars and nanoshells are the most promising (Figure 6). As expected, hybrid species with MS were successfully prepared. For a review dealing with Au@MS hybrid species, please check reference [127]. Although the use of Gold Nanostars (GNSts) in combination with MS is very recent, several examples with comparable efficiency to those obtained for the MSGNRs have been reported. In the first reported example using GNSts-MS hybrids, Zhang et al. designed a *Janus-*Au-Silica composite which was employed to successfully deliver DOX to HepG2 cancer cells [128]. Moreover, the *Janus* disposition of Au and Silica made it possible to functionalize both subunits differently: Au was capped with a stealth PEG polymer, while the silica moiety was decorated with lactobionic acid to improve its stability, biocompatibility, blood circulation time, and targeting towards cells bearing asialoglycoprotein receptors. In contrast to the following examples, which employ already shaped GNSts, in this example, the nanostar was prepared after the partial coating of an AuNP with MS. More recent examples dealing with GNSts can be found in the literature. For example, Martínez-Má$\tilde{n}$ez and coworkers developed an AuNSt@MS system coated with paraffin for DOX delivery [129]; while Raghavan et al. employed GNSts@MS for theranostic simultaneous PTE and photoacoustic detection [130]. In this last contribution, the authors obtained interesting results when a MS coated the nanostar: (1) a red-shift in the plasmon wavelength; and (2) an enhanced photoacoustic effect.

In addition to the strategy that employs Au embedded in a MS matrix, it is possible to carry out the opposite strategy, where Au particles are on the surface of regular MSNs. In addition to the expected greater sensitivity to stimulation with NIR, this strategy may simplify the gating process, as the AuNPs can act as nanogates [131]. This coating strategy was followed by Li et al. to develop themoresponsive nanodevices with multimodal imaging possibilities using GNSts [132]; demonstrating their adequacy for multimodally, ultrasonically, tomographically, photoacoustically and photothermally induced imaging. However, despite the promising future for these Au species, their in vivo behavior needs to be extensively evaluated.

In a parallel way, there is another family of Au-Silica hybrids that present a similar distribution of components: Gold Nanoshells (GNSs) [133]. These materials are prepared by coating a template with a thin Au layer, with silica being an outstanding platform for this due to its NIR transparency. Like Nanostars, GNSs can be employed for generating PTE, but not for the controlled delivery of drugs, as the template is sacrificed in a non-degradable coating. Like bare GNRs, nanoshells can be easily functionalized by employing the known reactivity of Au; however, contrary to the smaller particles, GNSs must be carefully employed in intravenous formulations, as their great size and highly difficult elimination will cause severe accumulation, and therefore long-lasting remainder effects. Nevertheless, these GNSs could be used in topical applications, as has been elegantly demonstrated by Mitragotri [134] (Figure 7) and Nie [135], who employed these materials to treat acne and melanoma, respectively. Moreover, PEG-coated silica-gold nanoshells have entered clinical trials for thermal ablation of solid primary and/or metastatic lung tumors (ClinicalTrials.gov identifier: NCT01679470), providing more arguments for silica to be used in nanomedical devices.

In addition to Au species, other degradable systems with plasmonic properties have been reported [136]. Among them, CuS has arisen as the most recurrent component among inorganic materials [118,119]. Like other PS, CuS has been successfully embedded into MS matrices to obtain functional photothermally active composites. In the first example reported, Song et al. prepared Cu_9_S_5_@mSiO_2_-PEG hybrids to treat Hep3B cells with simultaneous PTCT [137], demonstrating a comparable loading capacity to analogs containing GNRs. Additionally, hemolysis rates were similar to those obtained for PEG-coated MS, thus validating the suitability of these nanocomposites for use in combination therapy. Looking deeper into this topic, Zhu and coworkers employed an equivalent composite to successfully treat HeLa cells with a similar outcome [138].

In addition to PTE, CuS@MS composites have also been successfully employed in the development of materials for infrared thermal imaging and thermally responsive gated nanodevices. Along this line, Zhang et al. reported the construction of thermally triggered drug delivery nanodevices employing CuS@MS bearing aptamer-based nanogates [139]. The resulting system was able to address MCF-7 cells due to the targeting capabilities of the aptamer and to promote drug release when irradiated. For more information on the possibilities offered by DNA for developing mesopore gates and targeting, please check reference [140].

In addition to the use of MS as a coating component, CuS nanoparticles could also act as gatekeepers [141]. In this model, the authors employed a S-S cleavable linker to connect MSNs and CuS nanoparticles, enabling GSH redox-mediated cleavage of the system in intracellular environments. Viability studies showed cooperative apoptotic effects when the release of DOX was combined with the oxidative stress generated by GSH depletion and the induction of PTE throughout NIR irradiation.

In another interesting approach, Huang and coworkers prepared CuS crystals within the MSNs’ mesopores by employing thermal decomposition of Cu thiolates [142]. In a later step, the authors functionalized the outer MS layer with the Ir-2 PS, thus enabling dual photothermodynamic combination therapy. The system was activated when the typical radiations for each component’s (535 and 1064 nm) wavelength were employed, making it possible to destroy HeLa xenografts in mice in less than 7 days.

Another blooming field for CuS-containing composites is radiomedicine [143,144]. In a visionary example related to this topic, Cai and coworkers employed CuS@MS to create a traceable multifunctional nanodevice with targeting abilities. In order to do so, they decorated the outer MS layer with a TRC105 human/murine chimeric IgG1 monoclonal antibody and a ^64^Cu-chelated DOTA ligand [145]. The resulting system combined (1) highly effective targeting with (2) enhanced detection provided by the radioisotope and (3) the possibility of exerting PTE. This model, evaluated in a murine model, demonstrated complete tumor destruction when photoablation was exerted. 

In addition to the photothermal effect described for Au and CuS species, very recently, iron oxide nanoparticles have been reported to have a similar behavior [120,121,146]. This is of interest, because the dual mode of action makes it possible to access either deep tissues employing magnetic activation and surface tissues with light stimulation, together with the MRI contrast capabilities of Fe oxides. Unfortunately, despite the enormous potential of combining magnetic and light excitation with a single device, to the best of our knowledge, no studies on PTT in core-shell IOMSNs have been conducted, yet.

In summary, in light of the reported examples, it is clear that combined PTCT increases the effect of independent therapies. Nevertheless, these results must be treated with care, as the PTE itself is able to completely destroy tissues if enough radiation is applied. This effect, known as photoablation, is very useful for applying different degrees of tissue destruction, depending on the disease. Although it may be useful for treating highly resistant or difficult-to-operate tumors, it will always occur at the expense of completely destroying the tissues. Nevertheless, despite the huge potential of these materials, to date, no Au-based nanoparticles have been approved by the FDA, although it is claimed that barely modified Au nanorods and nanoshells are in the pipeline [1,2,147,148]. Under these circumstances, it is logical to assume that information on the the biosafety and biocompatibility of Au-MS hybrids will take several years to become generally available. Indeed, to advance in the use of Au for PTE, it will be necessary to solve several unanswered questions, such as the power and duration of irradiation necessary to achieve optimal therapeutic effect in different organs and tissues. However, there must also be information regarding the adequate thickness [149] and morphology [150] of the mesoporous matrix and its possible thermal insulating effect, and also regarding the toxicity [151], degradation and excretion processes related to nanomaterials for PTE.

### 3.3. Nanocomposites for Photodynamic Therapy

As outlined previously, PDE is a therapeutic effect based on the generation of ROS. However, in order to fulfill their function, it is important that ROS can effectively diffuse into the cytosol before self-destruction; therefore, their generation should occur at the outermost surface of the nanosystems. Hence, it is logical to assume that the classical core-shell architecture in which the PS occupies the central position is not the more convenient, as the ROS must get through the dense MS matrix. This circumstance has led researchers to become interested in surface decorated systems, which are easy to prepare by employing organic photosensitizers [107]. In addition to this, many researchers have also identified the generation of ROS as a side relaxation process after PTE generation. This could be employed to design multiple apoptosis-inducing systems with a single photon excitation. Interesting examples have been described with Au [152], CuS and CoS [153], although none have been described in combination with MS. For an outstanding and recent review on the modification of MSNs with different PS for PDE, please check references [154,155] (Figure 8).

### 3.4. Nanomaterials with Light Response: Fluorescence and Upconversion

Fluorescence and upconversion are two similar phenomena; both implicate the absorption of photons and their transformation into an emission of different wavelength. The difference between these two processes depends on the wavelength of the resulting emissions. Therefore, while fluorescence produces a lower energy radiation, upconversion transforms low-energy photons into high-energy radiation via a sequential excitation of the material throughout an anti-Stokes process. Obviously, for accomplishing upconversion, it is necessary to gather two or more photons.

As the outcome wavelengths are different for both processes, the application fields are different too. Fluorescence is the most important technique for detection of nanoparticles, being of interest for the detection and quantification of cell uptake and particle trafficking, fate and excretion [156]. Meanwhile, on the other hand, upconversion materials could be employed for either bright field detection (fluorescence) or photodynamic therapy, if the resulting radiation were able to trigger an apoptotic process or to excite a PS [107]. Herein, it is also important to remark that luminescence of UCNs is efficiently quenched by water, so the construction of composite materials needs to isolate the photoactive core from an intermediate dense silica layer.

Regarding the composition of these light-conversion materials, upconversion Nanoparticles (UCNs) are generally formed by fluorides of lanthanide trivalent elements (La^3+^, Gd^3+^, Y^3+^ and Yb^3+^), which have replaced some of the cations by dopant elements such as rare earth metals like Yb^3+^, Er^3+^, Tm^3+^, etc. [157,158,159]. On the other hand, excluding non-degradable C-dots [160,161,162], Quantum Dots (QDs) are mainly composed of nanocrystalline metal chalcogenides (Zn^2+^, Cd^2+^ or Pb^2+^ plus S^2−^ or Se^2−^), among many other minor compositions [163,164]. As could be figured out, the composition of these kinds of materials may have a relevant impact on the biosafety of both UCNs [165] and QDs [166,167], although this topic has still been only very poorly covered. In any case, it is logical to assume that their ionic nature would favor an eventual full degradation through the dissolution of crystals which, upon reaching a critical size, could be cleared out. However, their degradation occurs at the expense of releasing heavy metals [167,168,169,170], selenides and fluorides [171] into the organism. As a result, these nanocomposites must be very carefully applied, as they are formed by poorly biocompatible and toxic elements, which might expose the patient to a continuous dosage and undesired bioaccumulation. However, it is also important to note that the composite materials based on UCNs and QDs use very small crystals that could be quickly cleared upon degradation of the composites; although again, not much information is available on this topic.

The use of MS for the development of light-responsive composite nanomaterials makes it possible to incorporate a porous layer, which could be easily functionalized with polymers or targeting compounds and loaded with additional therapeutic compounds. Moreover, for most applications, this MS matrix is highly transparent, which ensures a proper response of the light-sensitive component, as long as the rest of the components do not interfere with the incoming radiation.

### 3.5. Quantum Dot Nanocomposites for Bright Field Detection

Fluorescence imaging is by far the most exploited tool for the detection, trafficking and bioaccumulation determination of nanoparticles in biomedical applications. Fluorophores can be either of organic nature, such as fluorescein and its derivatives, or inorganic nature, like quantum dots (QDs) and carbon dots [160]. Along this line, organic fluorophores have been successfully employed to prepare Cornell dots, which were the first in-human clinical trial for silica nanocomposites. These silica nanoparticles, loaded with the Cy5 fluorophore, functionalized with a PEG coating and a ^124^I radiolabeled cRGDY targeting peptide (ClinicalTrials.gov identifier: NCT01266096), are able to accomplish integrin recognition and hence to detect melanoma and brain tumors [172]. On the other hand, QDs offer advantages over the organic molecules, as they do not suffer from quenching and show higher quantum efficiencies with narrow adsorption and emission bands. The biocompatibility and the biomedical uses of QDs are highly dependent on their composition, which, besides containing heavy metals, also includes poorly biocompatible solvents employed in their synthesis [162,164,166,168,173]. This issue could be partially solved by coating QDs with a silica/MS layer able to displace the coordinated solvents while preserving [174] the QD core [175,176,177]. Unfortunately, the intrinsic toxicity associated with the forming elements and remainder solvents in their composition make them risky for in vivo biomedical applications. Nevertheless, their outstanding luminescence makes them ideal candidates for bioimaging and biosensors, as could be guessed by the overwhelming number of publications in these fields [178,179,180,181]. 

### 3.6. Mesoporous Silica Containing Upconversion Nanocomposites

Contrary to QDs, upconversion nanoparticles absorb low-energy photons and turn them into a higher-energy emission. This property, interesting for the development of nanoprobes and sensors [182], is also of interest for the development of therapeutic agents where the emission is energetic enough to perform cellular damage [183,184]. In order to achieve this, light upconversion is usually combined with photosensitizers.

In the first example of its kind, Idris et al. built a nanocomposite in which an Yb/Er-doped NaYF_4_ UCN was coated with a MS layer containing two PDE sensitizers: zinc phthalocyanine (ZnPC), and merocyanine 540 (MC540) [185]. This system was able to convert the incident 980 NIR radiation (2.5 W/cm^2^, 40 min) into visible radiation (550 and 660 nm respectively) able to independently activate each photosensitizer. The system was evaluated against B16-F0 melanoma tumors in a murine model, employing both naked or folate-PEG surface modifications. Under these circumstances, the targeted systems achieved a higher tumor growth disruption, albeit without tumor remission. More recently, Wu, He and coworkers improved the targeting ability of this system, employing a novel biomimetic camouflage strategy using stem-cell membranes [186]. The resulting upconverting protocells showed good stability and biocompatibility, together with a prolonged blood circulation time and the tumor-tropic properties of stem-cells. The triggering of the proapoptotic PDE with this nanodevice made it possible to significantly reduce the progression of HeLa tumors in mice, but only when the UCN@MS were coated. This result showed the huge potential of membrane-coated nanosystems for cancer treatment. Additionally, this strategy could also be employed to enhance long-term circulation and low immunogenic response, as elegantly reported by Xuan et al. [78], who cloaked their nanoparticles with red-blood cell membranes with outstanding results. 

The previous examples profited from double sensitizer photodynamic therapy in developing highly efficient cancer therapies; however, in those examples, mesopores were not occupied with any therapeutic agent. To cover this gap, Bu, Shi and coworkers developed a system for treating tumors with a synergistic reductive therapy based on the use of UCNs, together with a PS and a prodrug [187]. In order to do so, the authors assumed that upon PDE triggering (ROS generation), hypoxic environments are created. Therefore, if a prodrug that is activated under hypoxic conditions is co-administered, the apoptotic effect would be increased. The reported system employed Yb/Er/Gd-doped NaYF_4_ UCNs coated with silica as platform, silicon phthalocyanine dihydroxide as PS, and tirapazamine as prodrug. The potential therapeutic effect of the whole system on HeLa murine xenograft tumors provided promising tumor growth inhibition, despite it being a non-targeted nanosystem, demonstrating again the potential of combination therapy (Figure 9).

Additionally, UCNs@MS have been also successfully employed for drug delivery. In another contribution by Bu and Shi, an UV-Vis-triggered azobenzene-based nanoimpeller was employed to increase the outflow of DOX from the mesopores. In this case, the system was completed with the fusogenic TAT peptide as targeting element [188]. The authors claimed that the UCN was able to convert the incident NIR light (2.4 W/cm^2^) into a radiation able to photoisomerize the azobenzene, and hence to favor drug release acting as pore nanoimpellers. The in vitro evaluation of the resulting system on HeLa cells provided a nice correlation between irradiation time (5, 10, 15 and 20 min) and cell mortality. The authors associated the higher death rates to higher DOX releases. However, Dong and Zink addressed an in-depth study of the nanoimpeller-mediated release mechanism [189], finding that such enhanced DOX release did not occur due to photoisomerization. According to their investigations, the driving mechanism was a thermal effect created upon heating the UCN cores with the incoming 980 nm light, pointing out a PTE able to increase drug outflow, as demonstrated in light-induced drug release with UCN@MS in the absence of such nanoimpellers.

Another possible strategy to achieve PDE with UCN@MS composites is trapping the PS in the mesopores, as demonstrated by Han et al. [190]. In their system, the mesopores were loaded with Rose Bengal and capped with an adamantane–cyclodextrin nanogate. This approach, although it did not significantly improve the therapeutic potential of the other reported systems, opened the door to massive delivery of PS, which could be of interest for defining tumor burdens in post-treatment surgeries. 

Apart from upconversion of NIR into UV/Vis radiation, an interesting example was reported for the downconversion process, although in this case, the use of highly ionizing radiation significantly limits the applicability. In order to achieve this, Hirata’s group designed a Gd-Al garnet for X-ray downconversion [191]. The system was completed with a Rose Bengal photodynamic sensitizer that was able to transform the downconverted light into ROS. Regarding this nanosystem, it is noteworthy that the Gd-Al garnet enabled two unprecedented features for light-activated nanomaterials: (1) it enabled MRI detection; and (2) the downconversion of X-rays permitted an unprecedentedly in-depth tissue remote activation of nanosystems.

## 4. Multicomponent Nanocomposites Containing Mesoporous Silica

As already mentioned, the advantages of silica in the construction of nanomedical devices goes beyond its mere use as porous matrices. As reviewed, it also allows the quick and easy connection of functional entities. In the case at hand, the construction of multifunctional systems, there are many examples that use silica to integrate different types of particles and create nanosystems responsive to more than one remote stimulus. Among all published combinations, the most common is one that uses light and magnetic stimulation simultaneously. Following this strategy, Gao et al. reported the development of a multilayered, folate-targeted, Fe_3_O_4_@MS@CuS nanocomposite able to accomplish targeted PTCT and MRI [192], while Yao et al. reported the preparation of Fe_3_O_4_@MS@QDs device able to perform drug delivery and magneto-photothermal therapy with outstanding detectability [193].

In addition to the combination of SPIONs and AuNPs for magneto/photothermal therapy, this combination is also very useful for the preparation of multimodal detection systems, as the SPION enables MRI detection and Au does the same for X-rays computed tomography [194]. With this aim, Yang et al. prepared a multimodal detectable nanosystem in which a Fe_3_O_4_@Au composite was embedded into a MS matrix [195]. This system was further modified with three apoptotic inducers: a PS for PDE, DOX and a siRNA, which made it possible to obtain a tumor growth reversal in MCF-7/ADR cells. Another multipurpose assembly of AuNPs and SPIONs, reported by Sánchez et al., grew AuNPs onto IOMSNs. The resulting anisotropic Au-coated *Janus-*Fe_3_O_4_@MS composites [196] were able to perform MRI and computed tomography imaging. However, in addition to previous systems, the incorporation of a fluorescent Alexa Fluor 647 dye onto the exposed part of AuNPs made it possible to enable conventional bright field optical detection as well. Apart from Au, IOMSNs have also been combined with different inorganic species. For example, Cui and coworkers employed a silica-coated Fe_3_O_4_@ZnO composite to achieve microwave-triggered drug release. In this case, the ZnO interlayer acted as an ultrasound-to-temperature converter, albeit with poor efficiency [197].

In the same way, core-shell UCN@MS composites have also been the subject of additional modifications, seeking more versatile nanodevices for theranosis. Two examples with UCN cores were reported by Shi and coworkers, who designed a system that combined radio- and photothermal ablation by using ultrafine radioactive CuS nanoparticles deposited onto a UCN@SiO_2_ platform [198]; and by Liu et al., who successfully coated UCN@MS with SPIONs to accomplish bioimaging plus magnetically targeted DOX delivery [199]. Both examples made it possible to obtain great tumor destruction rates on murine models, demonstrating once again the potential of combining different functionalities into a nanodevice.

As reviewed so far, the combination of functional components in single nanometric formulations makes it possible to combine apoptotic effects, and hence to achieve better therapeutic profiles in the treatment of cancer. Unfortunately, to use these systems, a large number of biosafety limitations must be considered. These are not only consequence of the increasing complexity and number of connecting components, but also may be consequences of unforeseen side-effects that may arise when different functional components coexist in complex, living systems.

## 5. Conclusions

Despite raw mesoporous silica nanoparticles not having been approved as nanomedical formulations, the use of silica is widespread in the preparation of functional nanodevices. This is due to its extraordinary physicochemical behavior, biocompatibility, degradability and porous morphology. For this reason, materials formulated with silica in their composition have entered for the first time into the clinical studies pipeline, although there is still a long way to go before their use is generalized. 

From the data presented herein, it can be concluded that silica is no longer a material to be discounted by pharmaceutical companies. Indeed, it must be considered and properly studied as one more possible component of nanoformulations, since it provides a unique plasticity for the manufacturing of multifunctional devices. Above all, this must be tackled based on rigorous studies of acute and long-term toxicity, bioaccumulation and clearance of nanosized silica, in order to set a basis for the modelling of future composites. From a technical point of view, it is important to note that silica is a material that presents high permeation of (electro)magnetic fields and transparency against ultraviolet and infrared radiations, which is of interest for obtaining adequate remote activation of functional components. In short, in light of the published results, our opinion is that the use of silica in composite materials favors its translation into clinical phases, since unlike other components, the use of silica does not require a large number of connectors, complex synthesis processes, or poorly stable components.

## Figures and Tables

**Figure 1 ijms-20-00929-f001:**
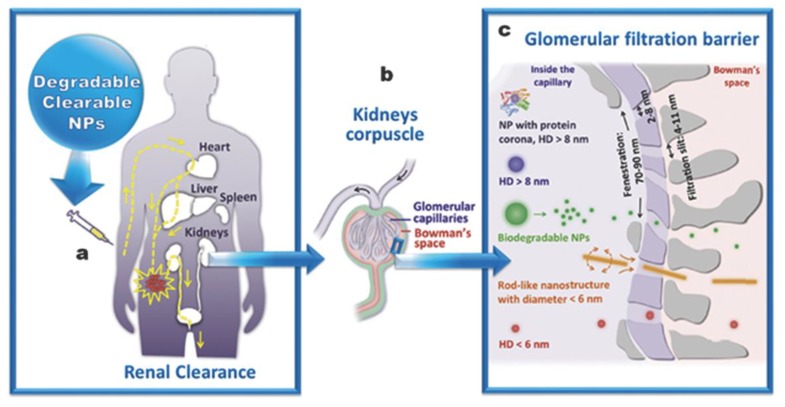
(**a**): Major organs involved in the biodistribution of nanoparticles. (**b**,**c**): Schematic glomerular clearance possibilities depending on the biodegradability and size of nanoparticles. Adapted from reference [34] with permission.

**Figure 2 ijms-20-00929-f002:**
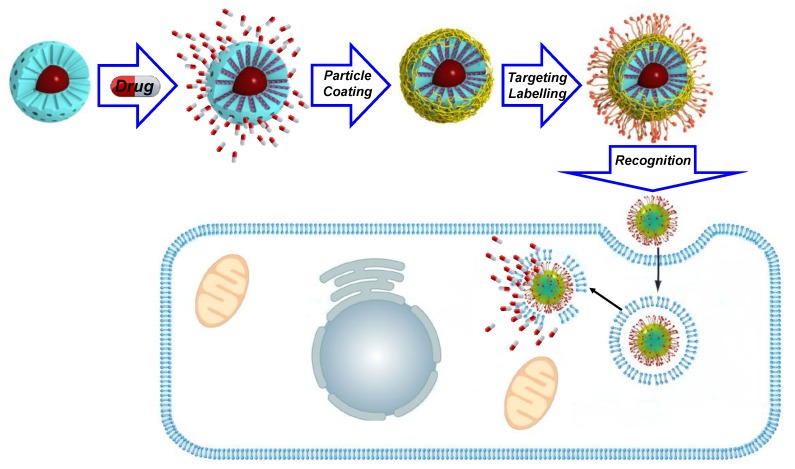
Typical strategy employed for the preparation of drug-loaded, polymer-coated, targeted IOMSNs reviewed herein.

**Figure 3 ijms-20-00929-f003:**
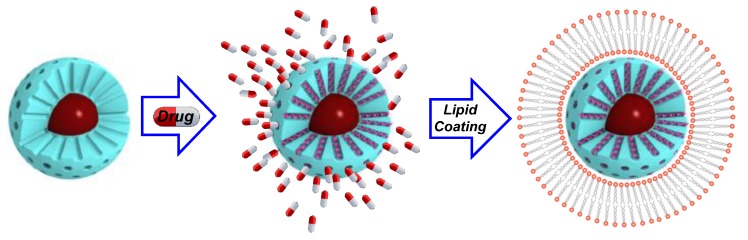
Multifunctional protocells arise from the coating of Mesoporous silica-based composites with lipid bilayers.

**Figure 4 ijms-20-00929-f004:**
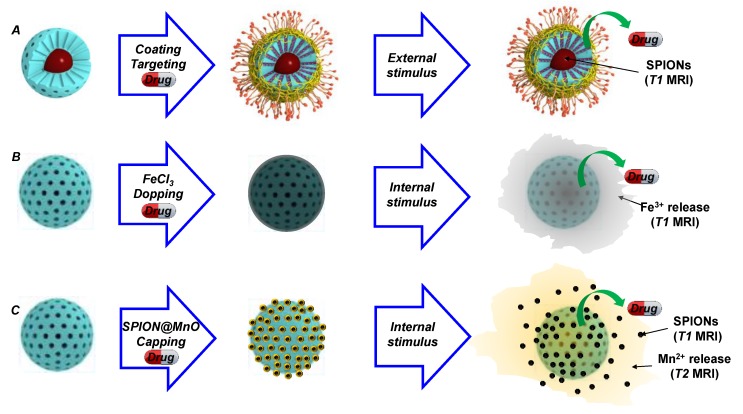
Possible strategies for the preparation of mesoporous silica-containing composites for MRI detection. (**A**) SPIONS contained in IOMSNs enable *T1* weighted MRI, while the outer mesoporous silica shell facilitates the development of hyperthermia-triggered systems. (**B**) Doping the silica matrix with acidic cleavable contrast cations for *T1* weighted MRI. (**C**) Capping mesopores with doped SPIONs for favoring acidic release of *T1* (SPIONs) and *T2* (Mn^2+^) contrast agents.

**Figure 5 ijms-20-00929-f005:**
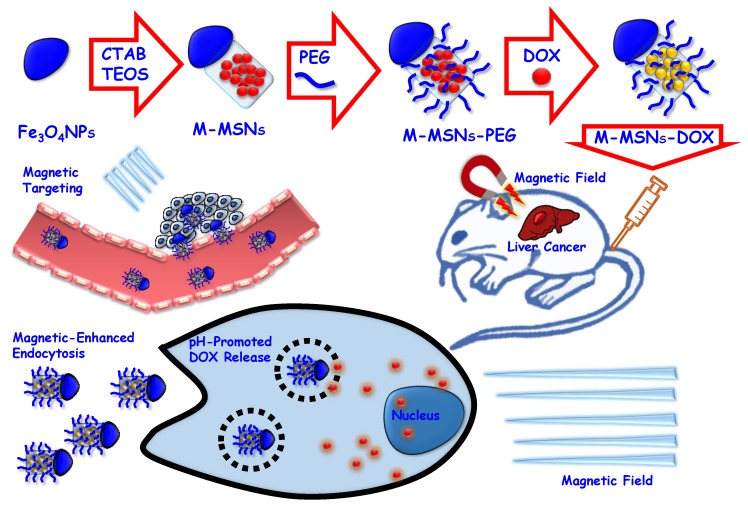
*Janus*-nanobullets employed for magnetic-targeted drug delivery.

**Figure 6 ijms-20-00929-f006:**
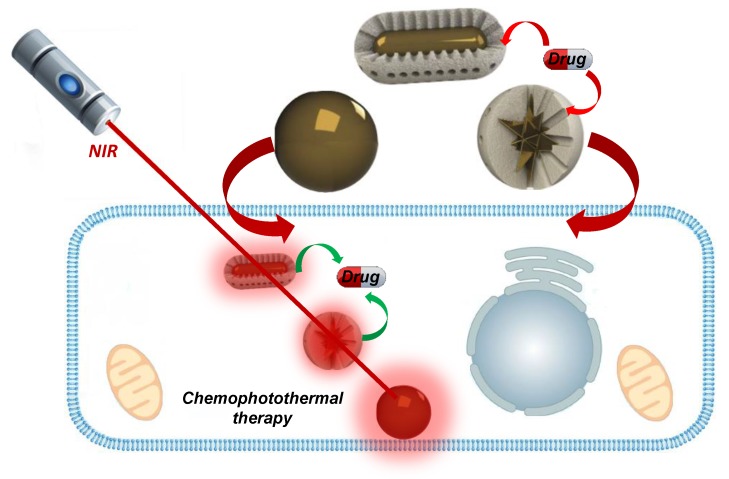
Au-Silica-based composites for photothermal applications. Gold nanorods, nanostars and nanoshells are able to transform near-infrared irradiation into thermal energy. The combination of such remote stimuli with drug delivery opens new possibilities for highly efficient therapeutic nanodevices. Unfortunately, the poor biodegradability of Au species makes their application poorly recommended for many treatments.

**Figure 7 ijms-20-00929-f007:**
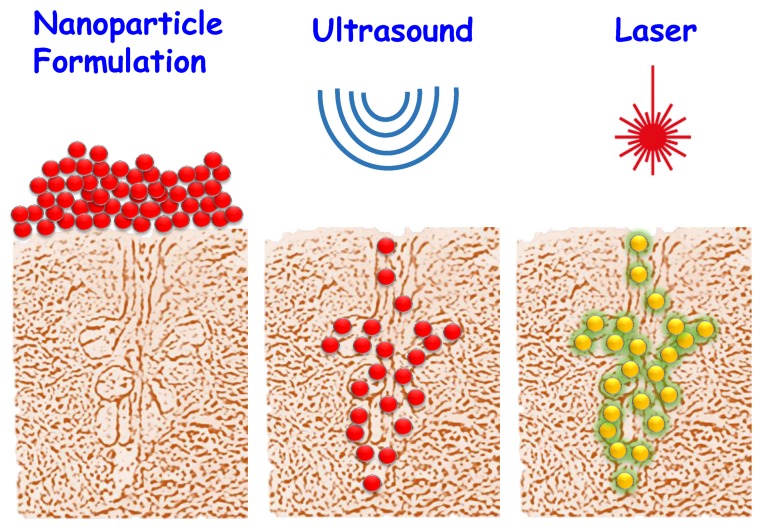
Two-step application of AuNSs for the treatment of acne. In this case, the topical administration of the AuNSs significantly reduces the risks associated with the “big” Au species.

**Figure 8 ijms-20-00929-f008:**
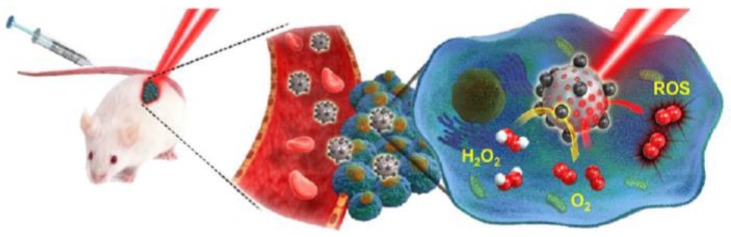
Photodynamic effect in therapy; the light stimulation is able to generate highly reactive oxygen species, inducing cellular apoptosis through oxidative stress and depletion of reductive species such as glutathione. The use of such dyes must be carefully addressed if they show long residence times, as their remainder activity may induce acute light sensitivity and chronic toxicity. Adapted from reference [155].

**Figure 9 ijms-20-00929-f009:**
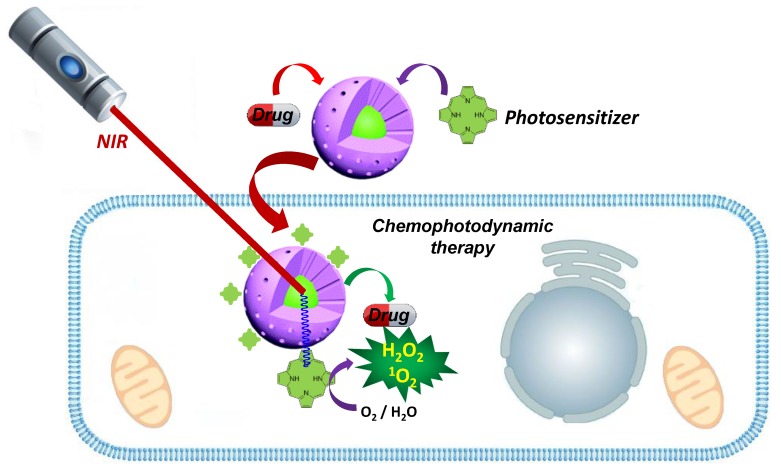
Upconversion-silica-based composites for photodynamic therapy. The internal core of these species is able to transform near infrared radiation into high energetic UV-Vis radiation suitable for exciting most photodynamic sensitizers known to generate oxidative stress-mediated apoptosis.

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
