# Peer review of "Functional Mesoporous Silica Nanocomposites: Biomedical Applications and Biosafety"

_ijms, 2019, doi:10.3390/ijms20040929_

Round 1

Reviewer 1 Report

Given the core topic of the review is mesoporous silica nanocomposites, it seems a couple of very basic issues have been overlooked. Mostly the text presents a collection of literature examples without regarding the basic concepts underlying them, which is why the text appears superficial (i.e. what can actually be learned here?) I will in the following point out a few things that should be added to improve this quite central shortcoming:

- given that mostly core-shell materials is in focus, also some kind of general description as to how mesoporous silica shells are synthesized could be given. This is not very straightforward and depends very much on the nature of the core material, so the general concepts in core-shell design could be introduced.

- by the same token, it is surprising to find that no notion has been given to the relaxivity enhancement that is usually observed when coating magnetic cores with MS shells. On the contrary, it has been stated that "However, the sensitivity provided by IOMSNs for T2-MRI is usually lower than those of SPIONs" which contradicts these observations. This is perhaps the most interesting property of IO-MS composites that can be attained by a MS coating.

- how is it possible that "The coating of SPIONs with silica is of interest because it allows to increase time-stability by preventing its dissolution" in case a MS coating is regarded? Here, another central aspect in core-shell design could be discussed which is that on occasion a "middle" non-porous silica coating may be needed (also for other reasons, such as separating the core IO from incorporated fluorescent dyes in the MS coating if designing magneto-fluorescent composites).

- Considering section 2.3 Magnetic Composites with applications in Nuclear Magnetic Imaging (MRI) how come Gd-chelated or Gd-doped MSNs have been omitted altogether?

- With regard to the same section, how is the release of different metal ions affecting the safety consideration (given "Biosafety" is one of the key words in the title)?

- In line with above, how are the release of pore capping agents (being inorganic materials of very small size) affect the safety considerations? How do caps vs cores compare here from an advantages/disadvantages point of view?

- The luminescence of UCNs are known to be very efficiently quenched by water. Here, a MS coating would not provide sufficient protection to circumvent this. This and considerations related to this has not been mentioned either.

- "Moreover, for most of applications this MS matrix is highly transparent, which ensures a proper response of the light sensitive component." This is a very simplified statement. While silica is known to be an optically transparent material, every added component to the design will affect the optical signal (e.g. drug loading, surface coating with organics etc.)

- Given all of the above, a discussion of the basic concepts underlying the design considerations would aid in shedding light on what kind of implications the addition of a MS coating (and further coating or funcitonaliztion) may have on any added functionality. These are very rarely purely "additive" in the way that e.g. the statement "To this end, many research groups employed already  developed core-shell systems for the preparation of complex composites" suggests. In the end, everything effects everything and this should be reflected in a topical review. With this in mind, the statment in the Conlusions section "which allow to maintain unaltered the properties of the functional components" is just plain wrong.

- It is also not correct to say "Although raw silica nanoparticles are not approved nor employed as a pharmaceutical formulations" given that 'Colloidal Silicon Dioxide' (characterized by a particle size of 15 nm) is a common excipient with many uses in many formulations.

Author Response

Given the core topic of the review is mesoporous silica nanocomposites, it seems a couple of very basic issues have been overlooked. Mostly the text presents a collection of literature examples without regarding the basic concepts underlying them, which is why the text appears superficial (i.e. what can actually be learned here?) I will in the following point out a few things that should be added to improve this quite central shortcoming:

First of all we want to thank the referee for all those valuable comments that surely will improve the overall quality of the manuscript. Particularly, we want to emphasize the comments done on catching our attention onto poorly or mistreated topics. We hope that all modifications made are in consonance with your suggestions.

- given that mostly core-shell materials is in focus, also some kind of general description as to how mesoporous silica shells are synthesized could be given. This is not very straightforward and depends very much on the nature of the core material, so the general concepts in core-shell design could be introduced.

Following your accurate recommendation we have included the following text on the manuscript: “Besides surface modification, it is also important to consider that connections between components require from additional in-between layers. One of the most common substances for such purposes is dense silica, usually employed in thicknesses around several nanometers. The role of this intermediate layer usually goes beyond providing chemical inertness and magneto-optical transparency. It also permits to physically separate both components, avoiding physicochemical processes like dissolution or passivation of the internal core and the quenching of fluorescent-labelled composites. Moreover, it also permits the generation of aditional mesoporous layers without adding complexity to the system. It is also important to account that these dense silica layers have demonstrated additional features, some of them reviewed below, as increased photothermal stability of cores or the possibility to tune relaxitivity of contrast agents in Magnetic Resonance Imaging [44-46].”

- by the same token, it is surprising to find that no notion has been given to the relaxivity enhancement that is usually observed when coating magnetic cores with MS shells. On the contrary, it has been stated that "However, the sensitivity provided by IOMSNs for T2-MRI is usually lower than those of SPIONs" which contradicts these observations. This is perhaps the most interesting property of IO-MS composites that can be attained by a MS coating.

Thank you for making us notice this mistake. Following your comment we have modified the manuscript as follows. “Moreover, the sensitivity provided by IOMSNs for T2-MRI is usually higher than those obtained for SPIONs, as recurrently demonstrated on the literature [83,84]. This effect could be also obtained by increasing the number of sensitizers throughout strategies that employ a greater number of metal oxides, which would be generally placed at the outermost layer of the composite.”

- how is it possible that "The coating of SPIONs with silica is of interest because it allows to increase time-stability by preventing its dissolution" in case a MS coating is regarded? Here, another central aspect in core-shell design could be discussed which is that on occasion a "middle" non-porous silica coating may be needed (also for other reasons, such as separating the core IO from incorporated fluorescent dyes in the MS coating if designing magneto-fluorescent composites).

True. We have simplified the statement as follows: “The coating of raw SPIONs is of interest because it allows to increase time-stability by preventing its dissolution and undesired aggregation processes, among other important features”. Regarding your second comment at this point, we now believe the intermediate dense silica layer role is clarified previously; so we did not make more additional comments at this point.

- Considering section 2.3 Magnetic Composites with applications in Nuclear Magnetic Imaging (MRI) how come Gd-chelated or Gd-doped MSNs have been omitted altogether?

Thank you for making us notice this missed information. We were focused on Fe-based compounds and completely skipped Gd. We have completed the chapter with the following paragraph dedicated to Gd-silica-based materials.

“Regarding Gd, the most widely employed contrast agent for MRI, there have been also reported a broad number of systems. Like in previous examples, the strongly paramagnetic Gd+3 ions could be either located onto the surface of MSNs throughout known chelants, doped within the porous silica matrix, or even as a core-shell structure. The first approach, chelation throughout ligands, proved to be suitable for the generation of MSNs with contrast properties for MRI [89,90]. Along this line, it is important to remark the work by Davis and coworkers, who determined that surface location of Gd-chelates led to better contrast and sensitivity [91]. Unfortunately, this approach has a severe drawback, as surface modification is highly limited and complex due to the presence of Gd-chelates.

For this reason new strategies have been developed for the incorporation of Gd into functional nanosystems. One of those possibilities is doping the silica matrix, which liberates the surface for further functionalization. However, a poor signal-to-noise ratio arises as a consequence of placing Gd in a mismatched crystallographic matrix. Hence, to overcome this issue two strategies that employ compact Gd-matrices has been reported. The first, in which the Gd occupies the core and the silica the shell, has the advantage of enabling multimodal detection throughout T1 weighted MRI and NIR-emitting persistent luminescence. Moreover, such approach allow to further modify the MS layer with all the developed targeting and stealthing technology, as elegantly demonstrated by Yu and Chen groups [92,93]. The other possibility for obtaining solid Gd-containing matrices is based on the construction of Gd-shells [94]; although this strategy does not profit from the advantages associated to mesoporous silica coatings..”

- With regard to the same section, how is the release of different metal ions affecting the safety consideration (given "Biosafety" is one of the key words in the title)?

About this topic, the previous version of the manuscript made remarks on that on chapters 2.3 (line 261) and 3.4 (lines 565 and on). Additionally, we have included such information on the paragraph that deals with the use of Gd for MRI. We believe that there is no need to specify more on the topic, however we would to include a remark somewhere if you consider it pertinent.

- In line with above, how are the release of pore capping agents (being inorganic materials of very small size) affect the safety considerations? How do caps vs cores compare here from an advantages/disadvantages point of view?

Thank you for this valuable comment. It is true that the advantages/disadvantages of each strategy are not specified on the manuscript. We have included the following paragraph on the end of chapter 2.3 to clarify it: “Regarding the biosafety of those species, there should be accounted that capping nanoparticles must comply with all the premises made at the beginning of this revision. They must have an adequate surface functionalization to provide enough colloidal stability and to avoid the action of the immune system. Moreover, nanocaps must be either efficiently cleared or biodegraded; although in any case, use of small nanoparticles for capping purposes clearly aid in their final elimination. Regarding the cores, the use of ionic species (such as iron oxides reviewed herein or CuS and UCNs reviewed below) do not seem to be a problem regarding bioaccumulation as they degrade; although they may suffer from acute toxicity issues. But the use of non-degradable species (as Au) must be carefully addressed, as will be discussed along the following lines.”

- The luminescence of UCNs are known to be very efficiently quenched by water. Here, a MS coating would not provide sufficient protection to circumvent this. This and considerations related to this has not been mentioned either.

Right. We made a comment on that at the chapter dealing with composites for upconversion. “Herein, it is also important to remark that luminescence of UCNs is efficiently quenched by water, so the construction of composite materials must need from an intermediate dense silica layer to isolate the photoactive core.”

- "Moreover, for most of applications this MS matrix is highly transparent, which ensures a proper response of the light sensitive component." This is a very simplified statement. While silica is known to be an optically transparent material, every added component to the design will affect the optical signal (e.g. drug loading, surface coating with organics etc.)

We agree with your point. We included a modification of such sentence for considering this accurate consideration. “Moreover, for most of applications this MS matrix is highly transparent, which ensures a proper response of the light sensitive component as long as the rest of components do not interfere with the incoming radiation.

- Given all of the above, a discussion of the basic concepts underlying the design considerations would aid in shedding light on what kind of implications the addition of a MS coating (and further coating or funcitonaliztion) may have on any added functionality. These are very rarely purely "additive" in the way that e.g. the statement "To this end, many research groups employed already  developed core-shell systems for the preparation of complex composites" suggests. In the end, everything effects everything and this should be reflected in a topical review. With this in mind, the statment in the Conlusions section "which allow to maintain unaltered the properties of the functional components" is just plain wrong.

We have taken into account this valuable comment and have modified both statements as follow:

Among all published combinations, the most common is one that uses light and magnetic stimulation simultaneously”

From a technical point of view, it is important to note that silica is a material that presents high permeation to (electro)magnetic fields and transparency against ultraviolet and infrared radiations, which is of interest for obtaining adequate remote activation of functional components”

Moreover, to deepen in the concept that everything affects everything, we have included the following paragraph at the end on the section.

“As reviewed so far, the combination of functional components in single nanometric formulations permits to combine apoptotic effects and hence to achieve better therapeutic profiles in the treatment of cancer. Unfortunately, to use these systems, there must be considered a large number of biosafety limitations. Those are not only consequence of the increasing complexity and number of connecting components, but also may be consequences of unforeseen side-effects that may arise when different functional components coexist in complex, living systems.”

- It is also not correct to say "Although raw silica nanoparticles are not approved nor employed as a pharmaceutical formulations" given that 'Colloidal Silicon Dioxide' (characterized by a particle size of 15 nm) is a common excipient with many uses in many formulations.

We agree that the sentence, as is, may be confusing. Our purpose is to state that nano-sized silica has no approval for drug delivery in intravenous formulations. Obviously, colloidal silica is widespread in many topical and oral formulations with a great controversy on its utilization. In any case we have rewritten the initial sentence as follows. “Although raw mesoporous silica nanoparticles have been not approved as a nanomedical formulations, the use of silica is widespread in the preparation of functional nanodevices.”

Reviewer 2 Report

Very pleasant Review! I suggest accepting it as it is!

Compliments to authors!

Author Response

Very pleasant Review! I suggest accepting it as it is!

Compliments to authors!

Thank you. We are very pleased by your support.

Reviewer 3 Report

The paper of Castillo and Vallet-Regí is interesting for people dealing with nanotechnology. The only issue is the absence of the use of IONs/mesoporous silica nanocomposites in photothermal therapy. For instance, New J Chem (2017), 49, 402-413. Other papers related can be found in the review of IONs in PTT (Molecules (2018), 23, 1567. 

Author Response

The paper of Castillo and Vallet-Regí is interesting for people dealing with nanotechnology. The only issue is the absence of the use of IONs/mesoporous silica nanocomposites in photothermal therapy. For instance, New J Chem (2017), 49, 402-413. Other papers related can be found in the review of IONs in PTT (Molecules (2018), 23, 1567.

We kindly appreciate the comment and we have included a paragraph on the manuscript including a reference on the topic. Nevertheless, we would like to remark that such topic was introduced previously (refs 121 and 122 of the corrected manuscript). Unfortunately we were unable to find your suggested reference: New J Chem (2017), 49, 402-413, and so we could not include it.

In addition to the photothermal effect described for Au and CuS species, very recently iron oxide nanoparticles have been reported to have a similar behavior [121,122,147]. This is of interest because the dual mode of action permits to access either deep tissues employing magnetic activation and surface tissues with light stimulation together with the MRI contrast capabilities of Fe oxides. Unfortunately, despite the enormous potential of combining magnetic and light excitation with a single device, up to our knowledge no studies on PTT in core-shell IOMSNs have been conducted yet.